# Beta HPV Deregulates Double-Strand Break Repair

**DOI:** 10.3390/v14050948

**Published:** 2022-04-30

**Authors:** Changkun Hu, Nicholas Wallace

**Affiliations:** Division of Biology, Kansas State University, Manhattan, KS 66506, USA; chu1@ksu.edu

**Keywords:** beta HPV, DSB repair, NHEJ, HR, Alt-EJ, MMEJ

## Abstract

Beta human papillomavirus (beta HPV) infections are common in adults. Certain types of beta HPVs are associated with nonmelanoma skin cancer (NMSC) in immunocompromised individuals. However, whether beta HPV infections promote NMSC in the immunocompetent population is unclear. They have been hypothesized to increase genomic instability stemming from ultraviolet light exposure by disrupting DNA damage responses. Implicit in this hypothesis is that the virus encodes one or more proteins that impair DNA repair signaling. Fluorescence-based reporters, next-generation sequencing, and animal models have been used to test this primarily in cells expressing beta HPV E6/E7. Of the two, beta HPV E6 appears to have the greatest ability to increase UV mutagenesis, by attenuating two major double-strand break (DSB) repair pathways, homologous recombination, and non-homologous end-joining. Here, we review this dysregulation of DSB repair and emerging approaches that can be used to further these efforts.

## 1. Introduction

The human papillomavirus (HPV) family is big, with over 450 HPV types already identified [1,2]. This virus family is subdivided into five genera (alpha, beta, gamma, nu, and mu) based on the sequence of the L1 capsid gene [3,4]. Alpha HPVs are most extensively studied, as a “high risk” subset of this genus can persistently infect mucosal epithelia leading to cervical, vulvar, vaginal, oropharyngeal, anal, and penile cancers [5,6,7,8]. Thanks to these efforts, life-saving vaccines against tumor-promoting HPVs have been developed and are widely available [6,9,10].

Unlike tumorigenic “high risk” alpha genus HPVs, beta genus HPV (beta HPV) infections occur in cutaneous epithelia [11,12]. In patients with a rare genetic disorder, epidermodysplasia verruciformis or EV, beta HPV infections persist and promote nonmelanoma skin cancer (NMSC) [13,14,15,16]. In EV patients, HPV5 and HPV8 are enriched in cutaneous squamous cell carcinoma (cSCC), which frequently occurs in sunlight exposed areas [16,17]. Infection with these viruses also increases the risk of cSCC in organ transplant recipients (OTRs) who are receiving immunosuppression treatment [4,5,6,7]. High beta HPV viral loads in OTRs result in >100-fold higher risk of cSCC [18,19]. 

Beta HPV infections are also common in the general population [20,21,22,23]. However, the fact that viral loads plummet to less than a copy per cell in cSCCs makes the extent to which they promote NMSC unclear [11,15,16,24]. It is clear however that beta HPV infections cannot promote tumors like known human oncogenic viruses, (i.e., by causing tumors that are dependent on continued viral oncogene expression). Further, higher beta HPV viral loads are found in precancerous skin lesions, such as actinic keratosis, but the amount of beta HPV dissipates to very low levels (<1 copy per cell) in cSCC [25,26]. It has been suggested that beta HPV increases mutations that drive tumorigenesis without continued viral gene expression being required for tumor maintenance or progression [25,27]. This would mean that beta HPV infections in the general population have their greatest impact on the cellular environment before transformation [15,28,29]. Epidemiological, animal, and cell culture systems support this hypothesis [15,16,25,30,31].

Here, we focus our discussion on the current understanding of how beta HPV hinders the repair of double-strand breaks in DNA (DSBs). The concentration on DSBs is based on the consensus that they are the most mutagenic type of DNA lesion. Further, failure to repair DSBs increases genomic instability and would be consistent with the hypothesized manner in which beta HPV infections promote NMSC formation. We also provide a survey of novel approaches that could be used to expand this knowledge base. 

## 2. Beta HPV and Genomic Instability

### 2.1. Beta HPV Attenuates the Cellular Response to UV-Induced Damages

Because beta HPV infections occur in the skin, ultraviolet radiation or UV is the most biologically relevant source of DNA lesions. The cellular response to UV-induced DNA damage is well characterized [32]. The most frequent lesions caused directly by UV are the cyclobutane pyrimidine dimers (CPDs) that form at two adjacent thymine bases [33]. CPDs stall replication forks/replicative polymerases, but do not impact replicative helicase activity. The uncoupling of polymerase from helicases generates a growing stretch of unstable single-stranded DNA (ssDNA). To prevent replication fork collapse, ssDNA is covered with an RPA trimer (RPA70, RPA32, and RPA14) [34]. With the help of TOPBP1 and ATRIP, the RPA trimer recruits ATR, the kinase that is largely responsible for coordinating the cellular response to stalled replication forks [35]. ATR activates itself by autophosphorylation [35]. Activated ATR phosphorylates downstream targets to initiate signaling cascades that turn on DNA repair and cell cycle arrest. Specifically, activated ATR phosphorylates CHK1 [32]. Phosphorated CHK1 phosphorylates and inactivates CDC25A and CDC25C, which are required for CDK2-mediated G1/S transition and CDK1-mediated G2/M transition, respectively [35]. ATR also activates a repair pathway known as nucleotide excision repair (NER) by phosphorylating XPA [36]. XPA abundance is rate-limiting for NER and its phosphorylation by ATR increases its stability [36]. This allows NER to remove UV lesions. ATR also facilitates translesion synthesis (TLS) by phosphorylating REV1 and DNA polymerase eta [35]. TLS is not directly responsible for repairing UV-lesions, but it helps prevent these lesions from causing replication fork collapse by allowing the replication fork to bypass these lesions. Finally, in response to UV damage, multiple kinases (ATR, CHK1, and casein kinase 2) induce further responses by phosphorylating and stabilizing p53 [32,35,37]. This leads to cell cycle arrest, apoptosis, and/or senescence.

High-risk alpha HPV encodes two primary oncogenes (E6 and E7) that facilitate extensive manipulation of the host cell environment. Studies of beta HPV biology often focus on the homologs of oncogenes [38]. This work primarily examines these homologs from only the subset of beta HPVs most closely linked to clinical manifestations (beta HPV5, 8, 20, 27, 38, and 49). Beta HPV studies have been performed in vitro using its natural host human keratinocytes. Human foreskin keratinocytes (HFKs) isolated from neonatal are frequently used because it has fewer mutations and are relatively easy to grow due to their young age. In human keratinocytes E6 from multiple beta HPV types contributes to the evasion of UV-induced apoptosis [39,40,41]. In HFKs, HPV5 and 8 E6 disrupt DNA repair pathways to make UV-induced CDPs more mutagenic [27,42,43,44]. HPV8 E6 leads to decreased ATR expression and activation in response to UV [42,43]. Further HPV8 E6 decreases downstream proteins of ATR in cells exposed to UV. This includes phosphorylated CHK1, phosphorylated XPA, total XPA, and total DNA polymerase eta (TLS) [43]. Moreover, HPV5 and 8 E6 bind to p300, a histone acetyltransferase that acetylates p53 in response to UV [42,45]. HPV38 attenuates p53 signaling and allows the proliferation of cells with UV damage [46]. 

The E6 and/or E7 from these beta HPVs also show transforming properties in vivo following UV radiation [47,48,49,50,51,52,53]. Particularly, HPV5 and 8 E7 expressing xenograft on humanized mice promote pretumor or tumorous skin lesions [51,54,55]. HPV8 E7 increases α3-integrin which promotes keratinocyte invasion [54]. HPV5 and 8 E7 also upregulate beta-catenin, which may contribute to the oncogenic potential of the virus [55]. Transgenic mice expressing HPV8 early genes develop cSCC, which can be enhanced by UV light [39,47,48,56,57]. Associated with these skin lesions, HPV8 E6 reduces phosphorylation of DNA damage sensing factors (ATR, CHK1, and ATM) [31]. HPV8 E6 and E7 work together to decrease CHK1 protein levels [58]. HPV8 complete early region (CER) also increases cancer-associated proteins including metalloproteinase, StefinA, and Sprr2 [59,60]. HPV8 CER also upregulates oncogenic miRNAs (17-5p, 21, and 106a) and downregulates tumor-suppressive miRNAs (155 and 206) [61]. Transgenic mice expressing HPV20, 27, 38, and 49 E6/E7 develop cSCC following UV radiation [29,43,49,52,53]. In support of the “hit and runs” mutagenesis hypothesis, deletion of E6/E7 after these lesions form did not affect cancer growth [29]. Rodent papillomaviruses resembling beta HPV were used to aid the study of cutaneous HPV biology in vivo [62,63,64,65]. Mastomys natalensis papilloma virus (MnPV) promotes cSCC in immunodeficient rats following UVB exposure [57]. Mouse papillomavirus type 1 (MmuPV1) induces papilloma in immunocompetent mice following UVB exposure [66].

### 2.2. Beta HPV Deregulates Double-Strand Break Repair Pathways

The interference with UV damage repair by HPV5 and HPV8 E6 causes a more frequent collapse of replication forks into mutagenic DSBs. Erroneous DSB repair can cause the loss of large regions of DNA, chromosome translocation/rearrangements, and aneuploidy [67,68]. Two major pathways, homologous recombination (HR) and non-homologous end-joining (NHEJ) have evolved to repair DSBs with high fidelity. In addition, alternative end-joining (Alt-EJ) serves as a more mutagenic backup DSB repair pathway should HR and/or NHEJ fail [69]. 

#### 2.2.1. Beta HPV Disrupts HR

Homologous recombination is an error-free DNA-repair pathway. It occurs in S/G2 phase so that a sister chromatid can be used as a homologous template. HR is dependent on DNA resection. ATM kinase and its targets are responsible for DSB sensing, cell cycle arrest, and DNA resection [70,71,72,73]. The MRE11, Rad50, and Nbs1 (MRN) complex together with CtIP to initiate minor DNA resection to generate 3′ short (~100 nt) ssDNA [74,75,76]. After short ssDNA is generated by the CtIP/MRN complex, downstream nucleases, and helicases such EXO1, DNA2, and BLM make extensive resection to reveal longer ssDNA [75,77]. With the help of mediator proteins BRCA1 and BRCA2, RPA and then RAD51 is recruited to the ssDNA to protect it from degradation [78,79,80,81,82]. RAD51-DNA filaments also facilitate homologous searching and strand invasion, which resolve the DSB [81,83]. 

A GFP-based HR reporter demonstrated that HPV5, 8, and 38 E6 decrease HR efficiency [84,85]. The mechanistic details of how this inhibition occurs have largely been worked out. By degrading transcription regulator p300, beta HPV 5 and 8 E6 decrease BRCA1 and BRCA2 at both mRNA and protein levels [84]. HPV5 and 8 E6 similarly reduce ATM protein abundance [84]. HPV8 E6 can also decrease ATM activation via phosphorylation [43]. Further, HPV5 and 8 E6 decrease the formation of BRCA1 and BRCA2 repair complexes. However, HPV5 and 8 E6 do not decrease RAD51 foci formation that is believed to depend on BRCA1 and BRCA2. Instead, they delay the resolution of RAD51 foci, suggesting that the repair complexes that form may be non-functional [84]. Supporting this idea, some of the RAD51 foci that form in cells expressing HPV8 E6 occur in G1 phase, when they are unlikely to be efficiently resolved due to the lack of a homologous template [86]. HPV38 E6 that weakly binds to p300 did not significantly decrease BRCA1 or BRCA2 expression. It also did not significantly alter HR repair complex formation or resolution.

#### 2.2.2. Beta HPV Attenuates NHEJ

NHEJ is responsible for DSB repair throughout interphase and G1 when homologous templates are not available. NHEJ initiates with localization of 53BP1 to the DSB. This helps prevent HR factors from promoting DNA resection [87,88,89]. Next, the Ku70/Ku80 dimer binds to DSB ends and recruits DNA-dependent protein kinase catalytic subunit (DNA-PKcs) [90,91,92]. Ku70/Ku80 and DNA-PKcs together form DNA-PK holoenzyme that facilitates NHEJ by phosphorylating downstream targets including Artemis [93,94,95]. Artemis has both endonuclease and exonuclease activity that helps to process the DNA end [93,95,96,97]. This end processing often includes the removal of overhangs and results in deletions. The resulting gap surrounding the blunt-ended DNA is resolved by XRCC4, XLF, and DNA ligase IV repair complex [94,98].

HPV5 and 8 E6 delay 53BP1 repair complex resolution [84]. Further, a CAS9-based NHEJ reporter showed that endogenous NHEJ was decreased in HPV8 E6 expressing cells [99,100]. The mechanistic analysis demonstrated that HPV8 E6 attenuated NHEJ by reducing the phosphorylation of DNA-PKcs and its downstream target Artemis. HPV8 E6 also impaired the resolution of pDNA-PKcs complexes to persistent [99]. These phenotypes were linked to HPV8 E6 mediated destabilization of p300. However, the viral protein decreased XRCC4 foci independent of p300 destabilization. These data confirmed that deletion of the p300 binding residues (amino acids 132–136) does not globally impair 8E6 activity. 

#### 2.2.3. Beta HPV Promotes Mutagenic DSB Repair Pathway

It should be noted that neither HPV5 E6 nor HPV8 E6 completely abrogates DSB repair. Rather, these viral proteins delay DSB repair. This suggests that DSB repair is still occurring and motivated ongoing efforts to identify the pathway(s) by which it was happening. To address this knowledge gap, a recent effort tracked the persistent HR and NHEJ foci found in earlier work [84,99]. This work found that HPV8 E6 caused HR factors (RPA70 and RAD51) to be recruited to sites of stalled NHEJ repair [86]. NHEJ and HR are intrinsically incompatible. HR requires ssDNA while NHEJ removes it. The colocalization of NHEJ and HR factors at the same DSB suggests that in cells expressing HPV8 E6, some DSBs are being repaired through an unusual, combined effort of the two pathways. This is expected to lead to increased mutations, especially deletions. Next-generation sequencing targeted at 200 kb surrounding a CAS9-induced DSB supports this idea, by showing a 20-fold increase in deletions in cells with HPV8 E6 compared to vector control cells [86]. Our unpublished data show that HPV8 E6 increases the use of a backup DSB repair pathway, known as alternative end-joining (Alt-EJ). Alt-EJ is intrinsically mutagenic and frequently detected in cancers [101]. Notably, Alt-EJ also results in deletions. These alterations in DSB repair suggest that HPV8 E6 makes DSBs more mutagenic. This may be a common property of cutaneous papillomaviruses as a recent study shows that MmuPV1 uses Alt-EJ to integrate DNA into the host genome in benign tumors [63].

## 3. Approaches to Study DSB Repair

### 3.1. Outstanding Questions

At least some beta HPV proteins abrogate correct DSB repair. However, outstanding questions remain. Many studies of beta HPV and DSB repair are conducted in cells expressing the E6 gene in isolation. While HPV8 E6 alone did not significantly decrease CHK1 protein levels, the combination of HPV8 E6 and E7 reduced CHK1 protein abundance in vitro and in vivo [58]. This suggests that HPV8 E7 augments HPV8 E6-mediated genome destabilization. To what extent, does co-expression of HPV8 E6 and E7 exacerbate the genome destabilization by HPV8 E6? More broadly, the most studied beta HPV proteins are HPV5, 8, 38, and 49 E6. Do other beta HPVs encode genes that hinder DSB repair? Do beta HPVs cause a unique enough pattern of mutations that they can be distinguished from mutations caused by other mutagens?

### 3.2. Induction of DSBs

DSBs can be induced by physical rays and chemical reagents. The source of these lesions is important to consider when studying DSB repair. UV and ionizing radiation (IR) are two types of radiation commonly used to generate DSBs in cells. Since UV relies on the collapse of replication forks, DSBs induced by UV do not occur in all cells at the same time [78,102]. Thus, the interpretation from kinetic studies becomes complicated. Moreover, UV especially UVA has low efficiency inducing DSBs [32,103]. Thus, only a fraction of cells will experience a DSB. In contrast, ionizing radiation (IR) efficiency induces DSBs by using high-energy particles to attack DNA strands directly [104,105]. This means that DSBs occur in nearly every cell at approximately the same time. However, IR also induces reactive oxygen species (ROS) that cause DNA damage [106,107]. As a result, interrogation of cellular responses may be complicated by the induction of signaling by ROS. ROS may complicate strict kinetic analysis as they independently cause DSBs. There are also practical considerations, for example, not all laboratories have access to sources of ionizing radiation. 

Chemicals can also be used to induce DSBs. For instance, radiation mimicking reagents such as bleomycin can cleave DNA via intercalation [108,109]. Hydrogen peroxide can also induce DSBs by increasing ROS [110]. The limitation of chemical reagents is similar to those associated with DSB-induction via UV, in that they complicate kinetic studies by inducing damage over time. One way of addressing this problem is to use a short pulse of media containing a high concentration of the drug of interest, but this approach should be taken with caution to assure that a subset of cells is not faced with non-physiological levels of DSBs. Additionally, most chemicals require vigorous washing to remove the compounds for any kinetic study of DSB repair. There is at least one exception. The radiomimetic drug, neocarzinostatin can be used to induce DSBs without the need for extensive washes as it becomes inactive (via degradation) within five minutes of treatment [111]. 

All the physical rays or chemical reagents mentioned in the preceding paragraph cause genome-wide DSBs in a largely non-discriminative manner. This increases the challenge of dissecting the repair process at a single DSB site. To study the repair of a single DSB repair at a targeted locus, specified artificial endonuclease such as I-Sce1 can be used [112]. However, these rare-cutting endonucleases require the integration of their recognition site into the genome of interest. Thus, the genome context is somewhat artificial. Further, integrating the I-SceI recognition site into the locus of interest and then validating its integration can be labor-intensive. The advent of sgRNA/CAS9 technology can subvert these restrictions allowing a DSB to be induced at a locus of interest without manipulation of the target cell beyond the transfection of the sgRNA and CAS9 expression plasmid [113].

IR and UV have been used to induce DSBs in cells with HPV5, 8, and 38 E6 [42,84]. Radiation mimic reagent zeocin was used to induce DSBs to study how HPV8 E6 deregulates NHEJ [99]. sgRNA/CAS9 was used to induce DSB at a single genomic locus in cells expressing HPV8 E6 [86,114]. Their ability to cause DSBs in different manners can be used to examine alterations in DSB repair caused by beta HPVs. For instance, UV can be used as a physiological source of DSBs. IR/zeocin can be used to directly induce DSB, so that differences in repair are not masked/exacerbated by upstream responses to UV that can make it more or less likely to cause DSBs. Similarly, zeocin, IR, and UV induce genome-wide DSBs, while sgRNA/CAS9 can be used to create a DSB at a specific site. While the random distribution of lesions is more physiologically relevant, inducing a DSB via sgRNA/CAS9 allows for the evaluation of the genome contexts of lesions to be more readily evaluated. 

### 3.3. Using Immunoblotting to Characterize DSB Repair Signaling

Activation of DNA repair is regulated by cellular signaling pathways [115,116,117]. Immunoblotting is a standard method to measure this activation in the form of altered protein abundance or post-translational modifications of signaling and repair factors. Phosphorylation and ubiquitination are common marks for the activation status of DNA repair proteins [118]. Immunoblotting together with densitometry can be used to measure the proportion of activated DNA repair proteins [99,119]. This is typically done following the induction of DSBs by one of the methods discussed in the preceding section. However, DNA repair is a complex, localized process. Higher DNA repair protein abundance (or post-translational modifications associated with repair factor activation) does not necessarily indicate higher DNA repair activity. DNA repair factors should bind to the damage site or interact with other repair factors [120]. Co-immunoprecipitation together with immunoblotting can be used to detect DNA repair factors physically interacting with known DNA binding proteins such as the DSB marker described in the next section [121]. Subcellular fractionation may similarly help define changes in localization indicative of DSB repair activation. 

Immunoblotting has been used to show that HPV5 and 8 E6 decrease major HR proteins including ATM, BRCA1, and BRCA2 [84]. The approach has also been used to show decreases in activation of NHEJ factors including pDNA-PKcs (S2056) and pArtemis (S516) [99]. Immunoblotting can be used to study how common these protein levels decrease in cells with beta HPVs or if these phenotypes change when E6 is expressed in the context of the whole viral genome or along with E7. Notably, HPV8 E6 decreases these DSB repair proteins above by destabilizing p300. Immunoblots can also be used to determine if HPV5 and 8 E6 make other DSB repair proteins are less abundant or less efficiently activated. Similarly, immunoblots can be used to determine if the expression of other beta HPV E6 genes decreases repair factor abundance. 

### 3.4. Immunofluorescence Microscopy of DSB Repair Factors in Fixed Cells

As described above, DNA repair proteins should be recruited to the damaged site forming repair complexes, visible by microscopy as distinct foci. Immunofluorescence (IF) microscopy is often used together with DSB induction to examine the localization of repair factors to DSBs. In these analyses, phosphorylated H2AX (S139) or pH2AX is an established marker of DSBs that can be used to confirm the localization of repair factors to DSBs or define the overall rate of DSB repair by fixing cells at intervals after DSB induction [122,123]. IF microscopy can also be used to characterize the kinetics by which other DSB repair complexes form and resolve, providing more detailed insight into repair [120]. Conjugated primary antibodies can be used to facilitate the detection of multiple protein targets in the sample. This can be used to detect the formation of multi-subunit repair complexes or combined with the detection of cyclin protein to determine when (regarding the cell cycle) repair complexes are forming [124,125,126]. 

IF microscopy has been used to detect repair kinetics of the HR pathway in cells expressing HPV5, 8, and 38 E6 [84]. The approach has also shown that HPV8 E6 delays the NHEJ pathway [99]. Future research can use IF microscopy to determine to what extent these changes in repair kinetics are induced by other beta HPVs and the impact of co-expressing E6 along with other beta HPV proteins. This approach can also be used to find more alterations in DSB repair kinetics, perhaps in other DSB repair pathways. 

### 3.5. Immunofluorescence Microscopy of DSB Repair Complexes in Living Cells

IF microscopy of fixed cells does not allow tracking of a single cell through the course of repair. Although technically more challenging, IF microscopy of living cells can address this barrier. Following DSB induction as described above, the recruitment and resolution of DSB repair factors can be recorded by time-lapse microscopy. This approach was used to reveal that NHEJ occurs in the G1 phase and NHEJ to HR pathway switch occurs in the S phase [87,127]. This approach has been used in combination with laser microirradiation to characterize the recruitment of repair complexes to laser-induced DSBs [128]. As a further tool, different types of lasers (UV, visible, near-infrared) can be used to induce different types of damage and to investigate different repair pathways [129]. Caution should be exercised with this approach as lasers can induce localized DSBs at unphysiological levels. Another challenge for imaging living cells is labeling DNA repair factors without altering their function. Typically, fluorescence tags are added to DNA repair proteins that are then transfected into cells. However, the increased abundance of certain DNA repair proteins may alter the DSB repair pathway choice. Further, the addition of the large fluorescence tag can impair protein function via steric hindrance [130]. One way of overcoming this challenge is to add smaller tags, (e.g., CLIP and SNAP) that self-label with fluorophores after repairing complex formation [131,132]. 

While live-cell imaging has been used to examine other aspects of beta HPV biology [46], it has not been used to study their impact on DSB repair. Thus, this approach has the most untapped potential of the techniques discussed in this review. One potential use would be to expand on our previous observation that HPV8 E6 promotes colocalization of NHEJ and HR factors [86]. Live-cell imaging would a definitive determination of how these abnormal complexes form and whether they are capable of resolving a DSB or marking cells for the death that is associated with an unrepaired DSB [133,134].

### 3.6. Reporters Constructs Can Measure Activity of Individual DSB Repair Mechanisms

To measure the frequency of a specific DSB repair pathway, specialized reporters have been developed. The general design of these reporters is that they contain a fluorescent protein (most often GFP) that is inactivated by insertions in the open reading frame [85]. I-SceI or sgRNA/CAS9 are used to induce DSBs [112,113]. Each reporter cassette is designed such that repair of the resulting DSB by a pathway of interest results in fluorophore expression. A recent adaptation of this approach uses CD4 expression as an alternative readout [100]. Using sgRNA/CAS9, two DSBs were induced at GAPDH and CD4 genes, if the DSBs are repaired by NHEJ it will result in deletion and rearrangement that will place the CD4 open reading frame just downstream of the GAPDH promoter. As a result, CD4 will be constitutively expressed. A limitation of this assay is that it can only be used in cells not already expressing CD4. However, because it does not use a reporter cassette, it has the advantage of measuring NHEJ frequency at DSBs occurring in the unaltered host genome [114]. Further, all of these systems rely on non-physiological relevant mechanisms of inducing DSBs, (e.g., I-SceI or sgRNA/CAS9), producing lesions that are often “cleaner” than naturally occurring DSBs and thus do not represent the complex nature of naturally occurring DSBs [114]. Finally, reporter assays can only detect the type of repair that they are designed for, thus variations that produce unexpected DNA products will not be seen. DSB repair reporters and measurements are summarized in Table 1.

**Table 1 viruses-14-00948-t001:** Reporters to measure specific DSB repair pathways. **Reporter** column lists common names for different DSB repair reporters. **DSB induction** column lists enzymes used to induce DSBs for each reporter. **Pathway and Readout** column lists the DSB repair pathway(s) measured by each reporter and the signal activated the pathway(s). **Reference** column lists the original publications where the reporters are described.

Reporter	DSB Induction	Pathway and Readout	References
DR-GFP	I-SceI	HR restores GFP	[85]
EJ2-GFP	I-SceI	Alt-EJ restores GFP	[135]
EJ5-GFP	I-SceI	NHE/Alt-EJ restores GFP	[135]
EJ7-GFP	sgRNA/CAS9	NHEJ restores GFP	[90]
4-μHOM	sgRNA/CAS9	Alt-EJ restores GFP	[90,136]
EJ-CD4	sgRNA/CAS9	NHEJ activates CD4	[100]

Reporter assays have been used to show that HPV5 and 8 E6 disrupt the HR pathway and that HPV8 E6 attenuates NHEJ [84,99]. Our unpublished data show that HPV8 E6 promotes Alt-EJ using the 4-μHOM reporter [90,136]. These reporters can be used to screen other beta HPV expressing cell lines for the ability to alter these DSB repair pathways as well as to determine the extent that which co-expression of E6 along with other beta HPV genes changed E6-mediated alterations in DSB repair.

### 3.7. Flow Cytometry

IF microscopy allows the DNA repair process to be studied at a high resolution, but this benefit is offset by the time-consuming nature of image capturing and analysis to obtain robust data. The reporter constructs described above tend to result in fluorophore expression at levels too low (<5%) to be amenable to detection by IF microscopy. Instead, flow cytometry is more commonly used as a high throughput alternative with these assays. It can be used to detect GFP from fluorophore-based reporter constructs. Flow cytometry can also be used to determine the cell cycle position of the repair complex and offer valuable insight into the interplay between cell cycle position and DSB repair factor activation [137]. With these approaches, cell cycle stages are most commonly determined by DNA content [138,139]. One weakness in this approach is that flow cytometry measures the total abundance or intensity of repair factors, which may or may not be equivalent to the detection of active repair complexes. Therefore, IF microscopy and flow cytometry should be used in tandem when investigating cell cycle-sensitive DNA repair factors. 

Flow cytometry is used together with specified DSB reporters to measure the efficiency of different repair pathways [84]. It also has been used to determine how HPV8 E6 alters the cell cycle distribution of RAD51 repair complexes [86]. Flow cytometry could similarly be used to determine the extent that which beta HPV proteins change the cell cycle distribution of other DSB repair complexes, or whether expression of the whole viral genome also leads to the same changes in the cell cycle distribution of DSB repair factors caused by HPV8 E6 expression. 

### 3.8. Next-Generation Sequencing

While the approaches above can be used to detect defects in DSB repair signaling, they do not measure the mutagenic impact of these defects (including increases in the number of changes in the types of mutations). Next-generation sequencing (NGS) is currently the best way to get this information. Whole-genome sequencing (WGS) is a type of NGS that offers an unbiased characterization of mutations by sequencing the entire genome. Whole exosome sequencing (WES) focuses this analysis by allowing mutations in non-protein-coding regions to be ignored. Targeted NGS can provide a robust analysis of a small region of interest allowing deep sequencing at an affordable price [140]. Targeted NGS can be paired with sgRNA/CAS9 technology to induce a single DSB at a defined locus and then obtain an in-depth analysis of the mutations associated with repair at that locus under experimental conditions of interest [86,114]. There are limitations to NGS approaches. For example, WGS can be cost-prohibitive and requires complex bioinformatics analysis. Perner et al. developed a sequencing strategy with reduced DNA quantity, which is more cost-effective than traditional WGS [141]. A random subset of the genome was obtained following double restriction enzyme digestion and size selection. However, this only yields data from areas where restriction ends are close enough together to facilitate sequencing and therefore be biased to certain regions. 

NGS at a known region (CD4) around a single DSB induced by sgRNA/CAS9 shows that HPV8 E6 increases multiple types of mutations [86,114]. NGS could also be used to examine how common mutagenic events are in cells with beta HPVs, whether expressing the whole viral genome leads to similar mutations induced by HPV8 E6, and how the pattern of mutations changes with preexisting mutations in the host genome. Similarly, interesting questions that could be addressed with NGS include whether the same pattern of mutations that are seen at CAS9-induced DSB occurs after UV or IR induced DSBs and how common it is for the E6 from other members of the beta HPV genus to cause increased mutations during DSB repair. Computational analysis of these data may allow a unique mutational signature for DSB repair in HPV8 E6 expressing cells to be identified, which might allow mutations promoted by HPV8 E6 to be distinguished from those caused independent of the virus. 

### 3.9. Approaches to Studying DSB Repair That Have Not Been Used to Study Beta HPVs

While most of the approaches described above have already been used to investigate beta HPV biology, their ability to ask other questions fundamental to cell biology allows them to remain useful. However, we are unaware of two of the approaches being used in this field, namely live imaging of repair complexes and WGS [86]. In addition to cell culture and rodent models, the organotypic raft can be a good model to simulate DNA damage response in differentiating skin [142]. While organotypic is commonly used in the investigation of alpha HPVs, these studies are less commonly applied to beta HPVs.

## 4. Summary and Discussions

Established and cutting-edge techniques have been used to show that beta HPV E6 proteins disrupt DSB repair and in some cases cause striking increases in mutagens during DSB repair. While these results support the “hit and run” hypothesis, they fall well short of providing definitive support for the idea. Ultimately, addressing outstanding questions (Section 3) is critical regardless of the outcome. Demonstrating a novel mechanism of tumorigenesis would be significant, but there is also significant value in a better understanding of a viral genus that infects most adults.

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
