# Peer review of "Beta HPV Deregulates Double-Strand Break Repair"

_viruses, 2022, doi:10.3390/v14050948_

Round 1

Reviewer 1 Report

The  authors reviewed beta human papillomavirus (beta-HPV) dysregulation of DSB repair and emerging approaches comprehensively and logically. This work addressed the knowledge gap in the mechanism of  beta-HPV infection in DSB and can help further investigation in beta-HPV.

However, the authors wrote in the abstract and introduction section” whether beta-HPV infections promote NMSC in the immunocompetent population is unclear. “ and didn’t answer or analyze this question. Please add relative contents or discussion.

PS:

In line 28-30 “In patients with a rare genetic disorder, epidermodysplasia verruciformis or EV beta-HPV infections persist and promote nonmelanoma skin cancer (NMSC) [10–13]”, the sentence should be “In patients with a rare genetic disorder, epidermodysplasia verruciformis or EV, beta-HPV infections persist and promote nonmelanoma skin cancer (NMSC) [10–13]”

Author Response

The authors reviewed beta human papillomavirus (beta-HPV) dysregulation of DSB repair and emerging approaches comprehensively and logically. This work addressed the knowledge gap in the mechanism of beta-HPV infection in DSB and can help further investigation in beta-HPV.

However, the authors wrote in the abstract and introduction section” whether beta-HPV infections promote NMSC in the immunocompetent population is unclear. “ and didn’t answer or analyze this question. Please add relative contents or discussion.

Thank you for this comment. Failure to repair DSBs can increase genomic instability and may promote NMSC formation. We now more clearly discuss this in line 49-51.

PS:

In line 28-30 “In patients with a rare genetic disorder, epidermodysplasia verruciformis or EV beta-HPV infections persist and promote nonmelanoma skin cancer (NMSC) [10–13]”, the sentence should be “In patients with a rare genetic disorder, epidermodysplasia verruciformis or EV, beta-HPV infections persist and promote nonmelanoma skin cancer (NMSC) [10–13]”

Thanks. This is corrected. Line 29.

Reviewer 2 Report

Drs Hu and Wallace set out to summarize the known oncogenic properties of betaHPV oncoproteins by focusing mainly on the activities that enhance the mutagenic effects of UV light. Before publication the following points need to be addressed:

  • General comment: Currently, the text is divided into two halves that are little connected. The first part describes the known oncogenic properties of betaHPV and the second part describes experimental methods that can be used to characterize DNA damage repair processes. However, these two parts need to be more closely interlinked. Even if the experimental approaches have not all been used to show the effect of betaHPV on DNA damage repair, one could describe the experimental set-up that can be used to answer the outstanding questions. By doing so, the rather simple table 2 would no longer be needed.
  • Line 21: The ref. 1 is quite old. Please cite here the PAVE (papillomavirus episteme) database.
  • Line 24: The authors need to include also oropharyngeal cancers as a high-risk induced cancer type and an appropriate reference for it.
  • Lines 35-44: The term “transient” is misleading here. Natural history analyses have shown that there are both persistent and transient infections of betaHPV on the healthy skin. Therefore, in this section, the authors have to describe "transient" precisely in order to avoid any misunderstanding.
  • Line 96: The reference for the HPV8-E7 mouse needs to be included (PMID: 26804167).
  • Lines 97-99: The reference PMID: 30786016 needs to be included as additional paper describing the loss of total Chk1 in HPV8-E6E7 positive keratinocytes. In addition, this reference can also be placed in line 330 where the authors discuss the role of E6 and E7 co-expression on the DNA damage repair pathway.

Author Response

  • General comment: Currently, the text is divided into two halves that are little connected. The first part describes the known oncogenic properties of beta-HPV and the second part describes experimental methods that can be used to characterize DNA damage repair processes. However, these two parts need to be more closely interlinked. Even if the experimental approaches have not all been used to show the effect of beta-HPV on DNA damage repair, one could describe the experimental set-up that can be used to answer the outstanding questions. By doing so, the rather simple table 2 would no longer be needed.

Thanks for your insight and suggestions. The revised manuscript lists outstanding questions before describing approaches. Following each approach, a summary was added to describe how the approaches can be used to answer outstanding questions. Table 2 was removed as suggested. Line 187-198 and line 238-406.

  • Line 21: The ref. 1 is quite old. Please cite here the PAVE (papillomavirus episteme) database.

Citations are updated. Line 20-21.

  • Line 24: The authors need to include also oropharyngeal cancers as a high-risk induced cancer type and an appropriate reference for it.

Oropharyngeal and anal cancer references are added. Line 24.

  • Lines 35-44: The term “transient” is misleading here. Natural history analyses have shown that there are both persistent and transient infections of beta-HPV on the healthy skin. Therefore, in this section, the authors have to describe "transient" precisely in order to avoid any misunderstanding.

Thanks for this comment. We revised these sentences to avoid misunderstandings. Line 35-39 and line 42.

  • Line 96: The reference for the HPV8-E7 mouse needs to be included (PMID: 26804167).

Thanks, this is added (ref 54). Line 97-98.

  • Lines 97-99: The reference PMID: 30786016 needs to be included as additional paper describing the loss of total Chk1 in HPV8-E6E7 positive keratinocytes. In addition, this reference can also be placed in line 330 where the authors discuss the role of E6 and E7 co-expression on the DNA damage repair pathway.

This is included in the revised manuscript and used as an example showing the importance of examining beta HPV genes in the context of the full viral early region. Line 102-103 and line 191-193.
